# Revisiting p53 Immunohistochemical Staining and Its Prognostic Implications in Advanced EGFR-Mutated Lung Adenocarcinoma

**DOI:** 10.3390/cancers17213577

**Published:** 2025-11-05

**Authors:** Feng-Che Kuan, Shun-Fu Chang, Yao-Ren Yang, Yu-Ying Wu, Fen-Fen Chen, Kam-Fai Lee, Chen-Lin Chi, Meng-Hung Lin, Chung-Sheng Shi

**Affiliations:** 1Graduate Institute of Clinical Medical Sciences, College of Medicine, Chang-Gung University, Taoyuan 333323, Taiwan; 8902029@cgmh.org.tw; 2Division of Hematology and Oncology, Department of Medicine, Chang-Gung Memorial Hospital, Chiayi 61363, Taiwan; 3Department of Medical Research and Development, Chang-Gung Memorial Hospital, Chiayi 61363, Taiwan; 4Center for General Education, Chang-Gung University of Science and Technology, Chiayi 61363, Taiwan; 5Department of Pathology, Far Eastern Memorial Hospital, New Taipei City 220216, Taiwan; 6Department of Pathology, Chang-Gung Memorial Hospital, Chiayi 61363, Taiwan; 7Health Information and Epidemiology Laboratory, Chang-Gung Memorial Hospital, Chiayi 61363, Taiwan; 8Department of Pulmonary and Critical Care Medicine, Chang-Gung Memorial Hospital, Chiayi 61363, Taiwan

**Keywords:** cancer, epidermal growth factor receptor, lung adenocarcinoma, Kaplan-Meier analysis, p53 protein, *TP53* mutation

## Abstract

**Simple Summary:**

The areas under the receiver operating characteristic (ROC) curves with different cut-off values for p53 positivity ranged between 0.51 and 0.56. Under a cut-off value of 50% p53 immunostaining, the sensitivity and specificity of *TP53* mutations were 66.7% and 45.5%, respectively, indicating an inconsistency between p53 immunostaining and *TP53* mutations. Tumor p53 overexpression (≥50%) was identified as a strong prognostic factor after adjusting for other important clinical factors, such as EGFR mutation subtypes, baseline brain metastasis status, and smoking status. This study shows that p53 immunohistostaining could help spare patients from toxicity of novel combination therapy once validated and adds value to *TP53* mutation analysis in the modern genomic era of EGFR-mutated lung adenocarcinoma.

**Abstract:**

Background/Objectives: *TP53* mutations in advanced epidermal growth factor receptor (EGFR)-mutated non-small cell lung cancer could worsen prognosis. Therefore, we aimed to investigate the clinical significance of *TP53* mutations and p53 expression in these patients. Methods: Patients with advanced/metastatic EGFR-mutated lung adenocarcinoma treated with first-line tyrosine kinase inhibitors were retrospectively enrolled. Sanger sequencing was performed to detect *TP53* mutations and immunohistochemical staining was used to verify p53 protein expression levels. Kaplan-Meier and Cox proportional hazards analyses were used to estimate survival and hazard ratio (HR) with 95% confidence interval (CI). Results: The study involved 83 patients with adequate tumor samples for *TP53*/p53 analysis. Patients with tumor p53 immunostaining ≥50% showed significantly better overall survival (OS) (HR: 0.49 [95% CI: 0.30–0.81], *p* < 0.001), but *TP53* mutations were not associated with inferior progression-free survival (PFS) or OS (missense vs. wild-type [PFS, HR: 0.68 (95% CI: 0.40–1.15), *p* = 0.151; OS, HR: 0.88 (95% CI: 0.56–1.42), *p* = 0.599]). Areas under the receiver operating characteristic curves of *TP53* mutations with different cut-off values for p53 positivity were 0.51–0.56. The Kaplan-Meier survival analysis revealed significant survival benefits in patients with EGFR L858R substitution and tumor p53 immunostaining ≥50% (median PFS: 8.0 vs. 5.3; median OS: 20.4 vs. 15.3 months; log-rank *p* = 0.025 and 0.049, respectively). Conclusions: Tumor p53 immunostaining (≥50%) was associated with better OS, especially in patients with *TP53* mutations or L858R. Prospective clinical trials are required to explore the prognostic significance of p53 expression in the genomic era of *TP53* mutations.

## 1. Introduction

The global incidence of lung cancer exceeded 2.5 million, accounting for 1.8 million deaths, in 2022 [1]. The development of therapies targeting biomarkers, such as epidermal growth factor receptor (EGFR)-activating mutations including deletions in exon 19 (Del19) and the single amino-acid substitution L858R in exon 21, has enabled personalized medicine for metastatic non-small cell lung cancer (NSCLC) [2]. Approximately 15–25% of Caucasians and 40–55% of East Asian patients with lung cancer carry EGFR mutations, and the evolution of tyrosine kinase inhibitors (TKIs) targeting EGFR mutations has improved progression-free survival (PFS) and overall survival (OS) in these patients [3,4,5]. However, the patients inevitably develop resistance to EGFR-targeted TKIs, at least partially because of the presence of other mutations along with EGFR mutations, which can influence and alter drug efficacy [6].

The human tumor suppressor gene *TP53* encodes p53, which transcriptionally controls the expression of numerous target genes involved in various biological functions. However, *TP53*, the most commonly mutated gene in human cancers, encodes defective p53 protein, which dysregulates transcriptional activities [7]. Furthermore, *TP53* mutations are among the most prevalent concomitant mutations in patients with EGFR-mutated lung cancer, and mounting evidence shows that *TP53* mutations can result in intrinsic resistance and poor prognosis [8,9,10]. A recent study reported that the presence of both *TP53* mutations and whole-genome doubling is associated with EGFR-TKI resistance [11]. A previous study has indicated that EGFR mutations and p53 overexpression play distinct roles in the development of small pulmonary adenocarcinomas [12].

Nevertheless, little is known about the relationship between p53 protein expression, as assayed through immunohistochemical staining, and the presence of *TP53* mutations, detected via sequencing, in patients with advanced EGFR-mutated NSCLC. Therefore, in this study, we aimed to investigate the association between these old and new approaches by verifying p53 protein expression, *TP53* mutations, and prognostic implications in patients with EGFR-mutated NSCLC.

## 2. Material and Methods

### 2.1. Patients and Tissue Samples

From January 2011 to June 2015, data of patients with stage IIIb/IV lung adenocarcinoma harboring EGFR mutations who were treated with frontline TKIs were retrieved from the medical records at Chang-Gung Memorial Hospital, Chia-Yi Branch, Taiwan. The following data were collected: sex, age (<70 and ≥70 years), smoking status, EGFR mutation subtypes (Del19, L858R, and other mutations), baseline brain metastasis status, and TKI types. Formalin-fixed, paraffin-embedded tissues of these patients were collected from the Chang-Gung Memorial Hospital Tissue Bank for p53 immunohistochemical staining and *TP53* mutation analysis. The need for patient consent was waived owing to the retrospective nature of the study. The study was approved by the Institutional Review Board of Chang-Gung Memorial Hospital (201900059B0).

### 2.2. p53 Protein Expression

Immunohistochemical staining of tissue sections was performed using a p53 monoclonal antibody (Glostrup, Denmark, Leica Biosystems; Cat# NCL-L-p53-DO7, RRID: AB_563936) as described previously [13,14]. The sections were counterstained with hematoxylin and eosin (Glostrup, Denmark) and blindly reviewed by two experienced pathologists (CFF and LKF) to identify areas with the highest tumor density (80% tumor content). Positive and negative controls were used to validate the reactions, and tumor samples with >10% of tumor cells showing positive nuclear staining were considered positive controls. p53 immunohistochemical staining was assessed semi-quantitatively based on the proportion and staining intensity of positive tumor cells (Appendix A and Appendix A). In cases of discordant p53 immunostaining results, a third pathologist (CCL) was consulted. None of the pathologists were aware of the molecular, biological, or clinical details of the patients.

### 2.3. TP53 Mutations

The high-resolution melting technique was used to quickly and efficiently screen for *TP53* mutations in exons 5–8, as previously described [15], using a 7900HT Fast Real-Time PCR System (Foster, Winston-Salem, NC, USA, Applied Biosystems). This was followed by Sanger sequencing (Appendix A). Mutations were classified as “disruptive” and “nondisruptive” according to the report by Molina-Vila et al. [15]. Also, in their analysis of the survival impact, *TP53* mutations were grouped as “wild-type + disruptive” and “nondisruptive.”

### 2.4. Statistical Analysis

PFS was calculated as described in our previous report [16] and OS was defined as the duration from TKI treatment initiation to patient death or loss to follow-up, whichever occurred first. Survival curves were generated using the Kaplan-Meier method, and the log-rank test was used to compare the time to events between the groups. Univariate and multiple regression analyses were performed using Cox proportional hazards models. Statistical significance was set at a two-sided *p* value of <0.05, and all analyses were conducted using SAS version 9.4 (SAS Inc., Cary, NC, USA).

## 3. Results

In total, 93 samples (including 2 paired samples) were obtained for analysis. After excluding 2 patients with incomplete medical information and 5 lacking sufficient tissue for p53 immunohistochemical staining and/or *TP53* mutation analysis, 83 patients with adequate formalin-fixed paraffin-embedded tissues for clinical and molecular analyses were enrolled. Of these patients, 57.8% were female individuals. The median age of the patients was 71 (range, 41–91) years; 66.3% of the patients were never smokers and 24.1% had brain metastasis at baseline. A total of 43 patients had the L858R mutation (51.8%), 25 patients had Del19 (30.1%), and 15 had other EGFR mutations (18.1%). Additionally, six patients received afatinib (7.2%), 21 received erlotinib (25.3%), and 56 received gefitinib (67.5%). A total of 39 patients (47%) had *TP53* missense mutations, of whom 26 had nondisruptive mutations (V216 deletions in exon 6 [*n* = 11] and R282L [*n* = 1] and R273L [*n* = 1] in exon 8) and 13 had disruptive mutations (H178R in exon 5 [*n* = 12] and R248E in exon 7 [*n* = 1]) (Table 1). Regarding p53 immunohistochemical staining, 45 patients (54.2%) showed 50% positivity, 63 patients (75.9%) showed >10% positivity, and 54 patients (65.1%) showed strong or intermediate intensity (Table 1 and Appendix A).

The median follow-up time was 20.3 months. The data showed that p53 immunohistochemical staining of ≥50% was associated with better OS in patients with *TP53* missense mutations in exons 5–8 (n = 39) (median OS: 23.8 vs. 19.6 months, log-rank *p* = 0.048, Figure 1A); however, this improvement in OS was not observed in patients with immunohistochemical staining of ≥10% (median OS: 20.4 vs. 22.9 months, log-rank *p* = 0.679, Figure 1D). When *TP53* missense mutations were subdivided into nondisruptive and disruptive mutations, there was only a trend toward improved survival in patients with nondisruptive mutations (log-rank *p* = 0.108, Figure 1C but not Figure B, E and F). Under a cut-off value of 10% p53 immunostaining, the sensitivity and specificity of *TP53* mutations determined through Sanger sequencing were 76.9% and 25%, respectively. Under a cut-off value of 50% p53 immunostaining, the sensitivity and specificity of *TP53* mutations were 66.7% and 45.5%, respectively (Appendix A), indicating an inconsistency between p53 immunostaining and *TP53* mutations in these samples. The areas under the receiver operating characteristic (ROC) curves with different cut-off values for p53 positivity ranged between 0.51 and 0.56 (Figure 2). No association was found between p53 immunohistochemical positivity (%) or p53 semi-quantitative staining intensity and *TP53* mutations (*p* = 0.413 and *p* = 0.526, respectively). Moreover, smoking status was not associated with *TP53* mutations (*p* = 0.695); however, p53 positivity (%) was associated with semi-quantitative staining intensity (*p* < 0.001).

In the univariate Cox regression analysis of PFS, older age (≥70 years) was associated with a better outcome (hazard ratio (HR): 0.58 [95% confidence interval (CI): 0.34–0.96], *p* = 0.035). Additionally, a trend toward better PFS was observed in patients with p53 positivity ≥50% (HR: 0.66 [95% CI: 0.40–1.08], *p* = 0.097), whereas patients harboring L858R showed a trend toward poor outcome (HR: 1.68 [95% CI: 0.95–2.97], *p* = 0.076). Moreover, regarding OS, ever/current smokers had a poor outcome (HR: 1.65 [95% CI: 1.00–2.70], *p* = 0.05) and p53 positivity ≥50% was associated with a better outcome (HR: 0.53 [95% CI: 0.32–0.84], *p* = 0.008). *TP53* mutations (nondisruptive vs. disruptive plus wild-type (WT) [PFS HR: 0.93 [95% CI: 0.54–1.58], *p* = 0.781 and OS HR: 0.85 [95% CI: 0.51–1.42], *p* = 0.531] or missense vs. WT [PFS HR: 0.76 [95% CI: 0.46–1.26], *p* = 0.284 and OS HR: 0.83 [95% CI: 0.52–1.33], *p* = 0.437]) and p53 semi-quantitative staining intensity (strong/intermediate vs. weak/negative [PFS HR: 0.90 [95% CI: 0.54–1.52], *p* = 0.696 and OS HR: 0.77 [95% CI: 0.48–1.26], *p* = 0.298]) were not associated with poor survival outcomes (Table 2).

In the multiple Cox regression analyses of PFS, older age (≥70 years) was associated with a significantly better outcome (HR: 0.50 [95% CI: 0.28–0.88], *p* = 0.017), whereas patients harboring L858R showed a trend toward poor outcome (HR: 1.81 [95% CI: 0.99–3.30], *p* = 0.053). In the multiple Cox regression analyses of OS, p53 positivity ≥50% was linked to a significantly better outcome (HR: 0.49 [95% CI: 0.30–0.81], *p* < 0.001), whereas baseline brain metastasis was associated with a poor outcome (HR: 1.84 [95% CI: 1.03–3.30], *p* = 0.041). However, *TP53* mutations (missense vs. WT [PFS HR: 0.68 [95% CI: 0.40–1.15], *p* = 0.151 and OS HR: 0.88 [95% CI: 0.56–1.42], *p* = 0.599]) were not associated with poor survival outcomes (Table 3). Additionally, *TP53* mutations (nondisruptive vs. disruptive plus WT [PFS 0.87 [95% CI: 0.50–1.52], *p* = 0.633 and OS 0.99 [95% CI: 0.58–1.69], *p* = 0.971]) were not associated with poor survival outcomes.

The Kaplan-Meier survival analysis of PFS showed a trend toward survival benefit in patients with p53 positivity ≥50% (median PFS: 11.3 vs. 7.3 months, log-rank *p* = 0.093). However, significant survival benefit was observed in patients with L858R and p53 positivity ≥50% (median PFS: 8.0 vs. 5.3 months, log-rank *p* = 0.025) (Figure 3A,C). A trend toward survival benefit was observed in patients with Del19 and p53 positivity ≥50% (median PFS: 12.9 vs. 11.1 months, log-rank *p* = 0.396) (Figure 3B). Additionally, a trend toward survival benefit was noted in patients with L858R and strong/intermediate p53 positivity (median PFS: 7.7 vs. 5.4 months, log-rank *p* = 0.077) (Appendix A). However, no differences in PFS were observed between patients with *TP53* missense mutations and those with WT (median PFS: 11.1 vs. 7.7 months, log-rank *p* = 0.285), or between patients with nondisruptive mutations and those with WT plus disruptive mutations (median PFS: 10.7 vs. 8.3 months, log-rank *p* = 0.786) (Appendix A).

The Kaplan-Meier survival analysis of OS showed a significant survival benefit in patients with p53 positivity ≥50% (median OS: 23.8 vs. 16.4 months, log-rank *p* = 0.007). Among these patients, those with L858R had a significant survival benefit (median OS: 20.4 vs. 15.3 months, log-rank *p* = 0.049), whereas those with Del19 only showed a trend toward survival benefit (median OS: 24.7 vs. 18.5 months, log-rank *p* = 0.121) (Figure 3E–G). There was no survival difference in patients with mutations other than Del19 or L858R (Figure 3D,H). No differences in OS were found between patients with *TP53* missense and those with WT (median OS: 21.6 vs. 19.3 months, log-rank *p* = 0.433) or between patients with nondisruptive mutations and those with WT plus disruptive mutations (median OS: 22.3 vs. 19.3 months, log-rank *p* = 0.529) (Appendix A). However, a significant OS benefit was observed in patients with Del19 and nondisruptive *TP53* mutations compared with that in patients with WT plus disruptive mutations (median OS: 32.7 vs. 19.3 months, log-rank *p* = 0.037) (Appendix A).

## 4. Discussion

To our knowledge, this is the first study to investigate the association between p53 expression and *TP53* mutations in patients with advanced EGFR-mutated lung adenocarcinoma. p53 immunostaining positivity of ≥50% was found to be a favorable prognostic factor for OS in the multivariate analysis of clinical factors, including EGFR mutation subtypes, baseline brain metastasis status, and smoking status [5,17,18]. Immunohistochemical methods to detect p53 were developed in the early 1990s, and because p53 is the “guardian of the genome,” it has been applied as a convenient surrogate for *TP53* mutation status to predict the survival outcomes in patients with cancer [19,20,21]. However, discrepancies between p53 expression levels and *TP53* mutations in lung cancer have been reported [22,23,24]. Over the past two decades, the importance of EGFR expression and mutations in lung cancer has been validated, and genomic medicine has led to improved treatments for patients with EGFR-mutated lung cancer [2,3,4,5,25,26]. Unexpectedly, p53 expression analysis using immunostaining has become largely obsolete in this field. We also observed a discordance in the present study findings when the monoclonal antibody DO-7 was used to detect p53 immunoreactivity. DO-7 can identify epitope mapping between amino acids 1 and 45 of human p53. It is recommended for the detection of both WT and mutant p53 proteins [20,27] under denaturing and non-denaturing conditions; thus, we could not confirm whether the higher expression of p53 was a WT or mutated phenotype. Furthermore, our immunohistochemical analysis data showed nuclear staining of p53 in all samples, suggesting that the transcriptional capacity of p53, which regulates genome integrity, might be preserved in these patients. Thus, a higher p53 immunoreactivity (≥50%) and nuclear localization staining might reflect the higher transcriptional activity of p53 in retaining its protective tumor growth-suppressive function. This finding is similar to that of a previous study, which showed that higher expression of p53 is a favorable prognostic factor in a subset of patients with NSCLC [28]. We found that when using a higher cut-off value of p53 immunostaining (≥50%), the rate of *TP53* missense mutations was 59.0%; it increased to 76.9% under a lower cut-off value of p53 immunostaining (≥10%). The enrichment of *TP53* missense mutations in patients with p53 staining ≥ 10% and the protective effect of p53 (≥50% staining) were attenuated. However, previous studies have focused solely on the prognostic implications of p53 protein expression levels or *TP53* mutations. We provide evidence that evaluating both p53 protein expression level and *TP53* mutations is important for predicting the outcome of NSCLC. Yoo et al. reported that EGFR mutations occur in the early stage and p53 overexpression is a late event in lung cancer development [12]. Our study further highlights that p53 positivity of ≥50% could identify a subgroup of patients with a more favorable prognosis in advanced EGFR-mutated lung adenocarcinoma.

Results for *TP53* co-mutations as a prognostic factor in patients with EGFR-mutated NSCLC have been inconsistent. A recent study concluded that the development of *TP53* co-mutations is a clinically relevant mechanism of EGFR-TKIs and a poor prognostic factor for OS [8]. However, in this study, there was no significant correlation between *TP53* co-mutations and PFS and OS, similar to the findings of other prospective and retrospective studies in patients with EGFR-mutated lung cancer [29,30,31,32]. This discrepancy in the findings may be because we focused only on *TP53* mutations from exons 5 to 8, consisting of the DNA-binding region, by direct sequencing of tumor tissues, which could have resulted in a low yield compared to that with next-generation sequencing of plasma samples [33]. Additionally, the *TP53* missense mutations in our study were mainly in exon 6 (74.4%, 29/39) and exon 8 (5.1%, 2/39), which differs from the findings of other retrospective studies [34,35,36]. A recent study has indicated that *TP53* mutations may have different effects on EGFR-dependent human malignancies [37]. In our study, a higher p53 protein level was associated with better clinical benefits in patients with EGFR-mutated lung adenocarcinoma. Mutated *TP53* may encode defective p53 proteins by abrogating the tumor growth-suppressive function; however, the presence of *TP53* mutations does not necessarily imply complete p53 inactivation. Mutant p53 may exert dominant negative effects by heterotetramerizing with WT p53 to interfere with its transcriptional activation. This may partially affect the binding of the mixed tetramer to DNA for the activation of some genes. Moreover, mutant p53 homotetramers may result in the loss of WT p53 function because of the inability to bind to p53 response elements and transactivate target genes. Furthermore, mutated p53 may confer various gain-of-function activities that affect cancer-related gene expression and possibly make the tumor more dependent on EGFR signaling [7,37], which, in turn, could make the tumor more sensitive to EGFR-TKIs.

In this study, nondisruptive *TP53* mutations, which can retain some functionality of the WT p53 protein, were associated with better OS in patients with Del19 (Appendix A). However, the incidence of nondisruptive mutations was lower (33% vs. >50%) than that observed in a previous study [15]. Moreover, combination therapy with vascular endothelial growth factor inhibitors or bispecific antibodies has been shown to potentially overcome the resistance to EGFR-TKIs associated with *TP53* co-mutations, but it is associated with toxicity [38,39,40]. In our study, the patients with L858R and p53 immunostaining ≥50% showed better PFS (8.0 vs. 5.3 months) and OS (20.4 vs. 15.3 months), which could spare these patients from toxicity. Although this study suggests the potential application of the traditional approach of p53 immunostaining to a modern approach of *TP53* mutation analysis using evolving sequencing methods, it has certain limitations. First, the study was retrospective in nature, with a small cohort of predominately non-smokers (66.3%). This further limits the power for subgroup analyses. Nevertheless, the pathologists (FFC, KFL, and CLC) were blinded to the molecular or clinical data of the patients, individuals managing the clinical data (YRY, YY, and MHL) were blinded to the molecular data, and the person dealing with the molecular data (SFC) was blinded to the clinical data. Further prospective studies are required to confirm and validate our findings. Second, we used only the monoclonal antibody DO-7 and the cut-off value for p53 immunostaining was arbitrary. Higher expression of p53, as detected using DO-7, was associated with better outcomes in a study by Lee et al. [28], but not in a study by Pollack et al. [20]. This discrepancy raises the question that our observations could be specific to lung cancer but not universal in different human malignancies. We propose a cut-off value of 50% with observed differences in OS, and this cut-off value should also be validated in a prospective study. Additionally, we used Sanger sequencing for *TP53* mutations spanning exons 5–8, which prevented us from analyzing their relationships with other important co-mutations using next-generation sequencing. Finally, the participants were enrolled from 2011 to 2015, and none of the patients received frontline osimertinib. These factors should be addressed in future studies to verify the complementary role of p53 immunostaining for individual *TP53* mutations.

## 5. Conclusions

In conclusion, this study suggests that p53 overexpression phenotype (≥50%) can identify a subgroup of patients with a more favorable prognosis, and this finding could potentially push the boundaries of genomic medicine for patients with EGFR-mutated lung adenocarcinoma with *TP53* co-mutations.

## Figures and Tables

**Figure 1 cancers-17-03577-f001:**
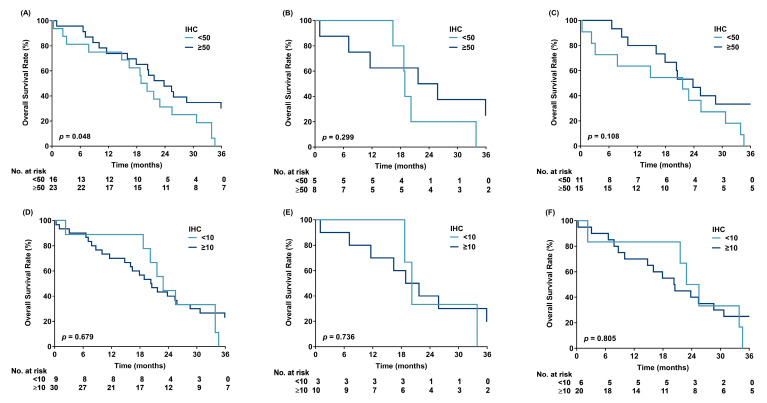
Overall survival based on p53 immunohistochemical positivity: (**A**–**C**) (≥50% vs. <50%) and (**D**–**F**) (≥10% vs. <10%). (**A**,**D**) Patients harboring missense mutations (MTs); (**B**,**E**) patients harboring disruptive MTs; (**C**,**F**) patients harboring nondisruptive MTs. IHC: immunohistochemical positivity.

**Figure 2 cancers-17-03577-f002:**
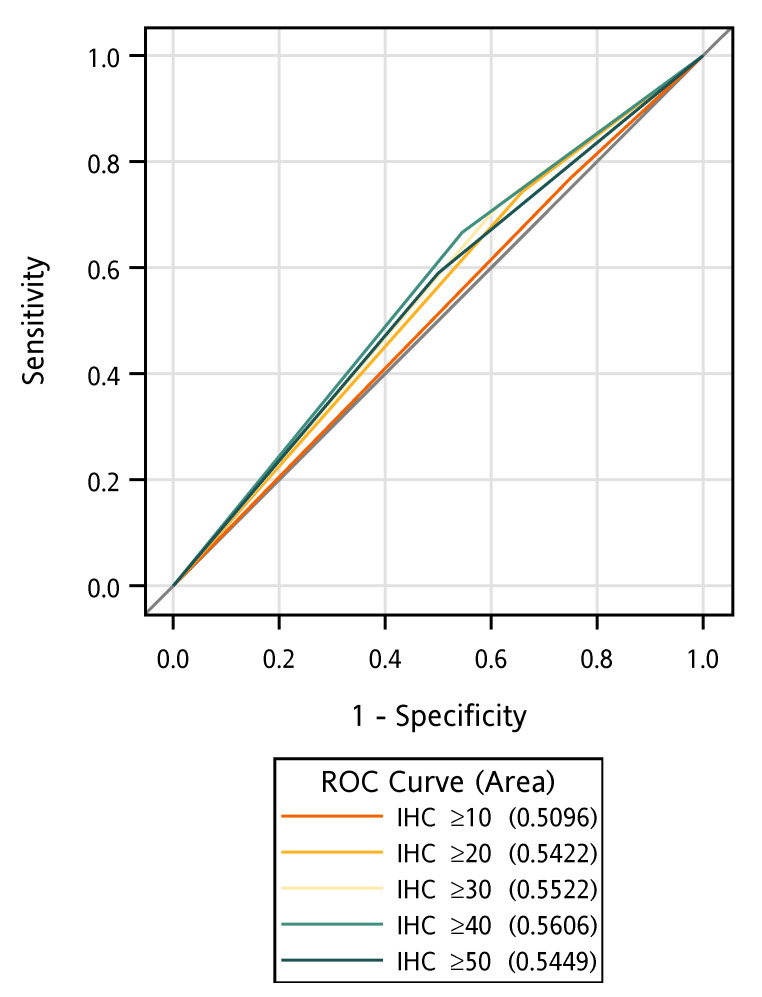
ROC curve by different cut-off values of p53 immunohistochemical positivity.

**Figure 3 cancers-17-03577-f003:**
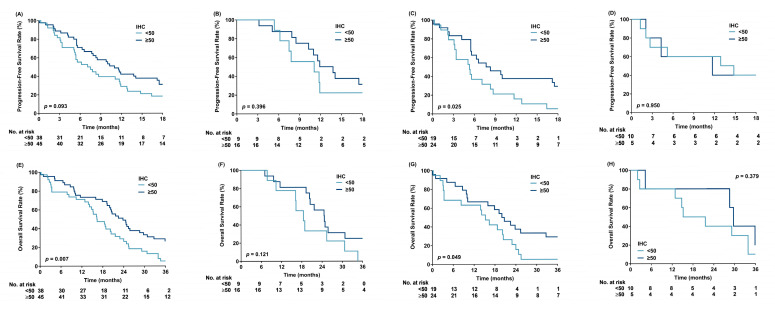
Progression-free survival (**A**–**D**) and overall survival (**E**–**H**) based on p53 immunohistochemical positivity (≥50% vs. <50%). (**A**,**E**) All patients; (**B**,**F**) patients harboring Del19; (**C**,**G**) patients harboring L858R; and (**D**,**H**) patients harboring other mutations.

**Table 1 cancers-17-03577-t001:** Baseline characteristics of the patients.

Variable	*n*	(%)
Total	83	
Sex		
Female	48	(57.8)
Male	35	(42.2)
Age (years)		
<70	41	(49.4)
≥70	42	(50.6)
Median (range)	71	(41–91)
Smoking		
Never	55	(66.3)
Ever/Current	28	(33.7)
Baseline brain metastasis		
No	63	(75.9)
Yes	20	(24.1)
EGFR mutations		
Del19	25	(30.1)
L858R	43	(51.8)
Other	15	(18.1)
p53 positivity		
<50	38	(45.8)
≥50	45	(54.2)
p53 mutation		
Nondisruptive MT *	26	(31.3)
WT + Disruptive MT	57	(68.7)
p53 mutation		
Missense	39	(47.0)
WT	44	(53.0)
p53 intensity		
Strong/Intermediate	54	(65.1)
Weak/Neg	29	(34.9)
TKIs		
Afatinib	6	(7.2)
Erlotinib	21	(25.3)
Gefitinib	56	(67.5)

* The MT refers to *TP53* mutation.

**Table 2 cancers-17-03577-t002:** Univariate Cox regression analysis of progression-free survival and overall survival in patients with advanced EGFR-mutated adenocarcinoma.

	PFS		*p*-Value	OS		*p*-Value
Variable	HR	(95% CI)		HR	(95% CI)	
Sex						
Female	Reference			Reference		
Male	0.85	(0.51–1.40)	0.517	1.32	(0.82–2.14)	0.257
Age (years)						
<70	Reference			Reference		
≥70	0.58	(0.34–0.96)	0.035	1.19	(0.74–1.90)	0.477
Smoking						
Never	Reference			Reference		
Ever/Current	1.01	(0.65–1.85)	0.722	1.65	(1.00–2.70)	0.050
Baseline brain metastasis						
No	Reference			Reference		
Yes	1.25	(0.71–2.18)	0.440	1.55	(0.90–2.66)	0.117
EGFR mutations						
Del19	Reference			Reference		
L858R	1.68	(0.95–2.97)	0.076	1.24	(0.72–2.13)	0.438
Other	0.94	(0.42–2.09)	0.871	0.98	(0.49–1.97)	0.961
p53 positivity						
<50	Reference			Reference		
≥50	0.66	(0.40–1.08)	0.097	0.53	(0.32–0.84)	0.008
p53 mutation						
Nondisruptive MT	0.93	(0.54–1.58)	0.781	0.85	(0.51–1.42)	0.531
WT + Disruptive MT	Reference			Reference		
p53 mutation						
Missense	0.76	(0.46–1.26)	0.284	0.83	(0.52–1.33)	0.437
WT	Reference			Reference		
p53 intensity						
Strong/Intermediate	0.90	(0.54–1.52)	0.696	0.77	(0.48–1.26)	0.298
Weak/Neg	Reference			Reference		
TKIs						
Afatinib	0.62	(0.19–2.01)	0.426	0.65	(0.23–1.80)	0.404
Erlotinib	0.85	(0.48–1.52)	0.587	0.83	(0.48–1.43)	0.509
Gefitinib	Reference			Reference		

**Table 3 cancers-17-03577-t003:** Multiple Cox regression analysis of progression-free survival and overall survival in patients with advanced EGFR-mutated adenocarcinoma.

	PFS		*p*-Value	OS		*p*-Value
Variable	HR_adj. _*	(95% CI)		HR_adj._	(95% CI)	
Sex						
Female	Reference			Reference		
Male	0.40	(0.15–1.07)	0.068	0.84	(0.33–2.11)	0.707
Age (years)						
<70	Reference			Reference		
≥70	0.50	(0.28–0.88)	0.017	1.59	(0.93–2.72)	0.088
Smoking						
Never	Reference			Reference		
Ever/Current	2.11	(0.77–5.81)	0.148	1.94	(0.76–4.93)	0.165
Baseline brain metastasis						
No	Reference			Reference		
Yes	1.23	(0.68–2.22)	0.495	1.84	(1.03–3.30)	0.041
EGFR mutations						
Del19	Reference			Reference		
L858R	1.81	(0.99–3.30)	0.053	1.15	(0.66–2.01)	0.623
Other	1.08	(0.43–2.71)	0.878	0.74	(0.34–1.59)	0.441
p53 positivity						
<50	Reference			Reference		
≥50	0.70	(0.41–1.20)	0.194	0.49	(0.30–0.81)	<0.001
p53 mutation						
Missense	0.68	(0.40–1.15)	0.151	0.88	(0.56–1.42)	0.599
WT	Reference			Reference		

* The HR_adi_ refers to the hazard ratio adjusted by sex, age, smoking, baseline brain metastasis, EGFR mutations, p53 positivity and *TP53* mutation.

## Data Availability

The raw data supporting the conclusions of this article will be made available by the authors on request.

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
