# Peer review of "Revisiting p53 Immunohistochemical Staining and Its Prognostic Implications in Advanced EGFR-Mutated Lung Adenocarcinoma"

_cancers, 2025, doi:10.3390/cancers17213577_

Round 1

Reviewer 1 Report

Comments and Suggestions for Authors

Major Concerns

  1. The study only examined TP53 exons 5–8, using Sanger sequencing rather than NGS. Given the widespread distribution of TP53 mutation hotspots (particularly in regions outside the DNA-binding domain that also have functional implications), this detection strategy may underestimate the heterogeneity in mutation frequency and type. It is recommended that this methodological limitation be more clearly stated in the Discussion to avoid over-extrapolating the conclusion that TP53 mutations have no prognostic significance. If possible, additional analysis of other mutation sites or comparison with NGS studies should be included. Furthermore, the high missense mutation rate of 39/83 (47%) in the paper's results may be due to sample selection bias, and further validation requires whole-exome NGS.
  2. The scientific basis for the selection of the p53 immunohistochemistry cut-off is insufficient. The study used 10% and 50% cut-offs, but the rationale for these two selections was not fully explained. The prognostic significance of the 50% cut-off may be due in part to statistical fluctuations rather than biological mechanisms. Authors are advised to provide more rationale for the selection of the ROC analysis threshold or cite more supporting literature.
  3. The article mentions an AUC of only 0.51–0.56, indicating poor concordance between immunohistochemistry and TP53 mutation. However, the discussion primarily emphasizes the "protective" significance of high p53 expression without fully explaining the biological basis for this discordance. For example, wild-type p53 is upregulated by stress; the DO-7 antibody may recognize both WT and mutant proteins; and certain missense mutations may result in a stabilized but inactive p53 protein. A more in-depth discussion of these discrepancies is recommended, as otherwise the conclusions may appear overly simplistic.
  4. Statistical analysis and biological interpretation may be confounded. Although Cox regression results showed that p53 ≥ 50% was associated with better OS, no significant difference was found for TP53 mutations, suggesting the possibility of confounding factors (such as EGFR mutation subtype, treatment era, and other concomitant mutations). The limited inclusion of variables in the multivariable model may not fully adjust for confounding factors. Authors are advised to perform a sensitivity analysis or be more cautious in interpreting causal relationships.

Minor Concerns

  1. The methodological description is not standardized enough. The IHC criteria are simply described as "strong/intermediate/weak." It is recommended to further clarify the scoring system used or adopt a more quantitative method such as the H-score. Additionally, the manuscript would benefit from including representative immunohistochemistry (IHC) images demonstrating the staining patterns used to define “weak,” “intermediate,” and “strong” expression, as well as the chosen cut-off values (e.g., <10%, 10–50%, ≥50%).
  2. Several statistical results are marginally significant, such as the p=0.097 for p53 ≥ 50% in PFS. Simply referring to it as a "trend" is more appropriate. The main text should be more cautious in its description to avoid misleading information.
  3. The biological background of the DO-7 antibody recognition spectrum and the functional differences between TP53 mutation types (nondisruptive vs. disruptive) should be supported by more authoritative reviews or TCGA data.
  4. Subgroup analysis figures (e.g., Figure 3) are best labeled to indicate which subgroup they represent. HRs can also be annotated to enhance the information.
  5. Some paragraphs are too long (especially the Discussion). It is recommended to break them up into sections and refine the key points. Using "suggest" instead of "confirm" in some sentences would be more consistent with the scientific tone.

Reviewer 2 Report

Comments and Suggestions for Authors

The authors reported their manuscript named "Revisiting p53 Immunohistochemical Staining and Its Prognostic Implications in Advanced EGFR-Mutated Lung Adenocarcinoma.". This is a well-written and clinically relevant retrospective study that investigates the complex relationship between p53 protein expression by immunohistochemistry (IHC) and TP53 mutation status in a cohort of advanced EGFR-mutated lung adenocarcinoma patients. The central finding—that high p53 IHC expression (≥50%) is an independent favorable prognostic factor for overall survival (OS), particularly in specific subgroups, while TP53 mutations themselves were not prognostic—is intriguing and counter to some prevailing assumptions. The study addresses an important gap in the era of genomic medicine by revisiting the utility of a traditional biomarker (p53 IHC). The manuscript is generally clear, the methodology is sound for the techniques used, and the statistical analysis is appropriate. I have the following comments:

Major Points

  1. Interpretation of p53 IHC and Its Discordance with Mutation Status: The manuscript correctly identifies a significant discordance between p53 IHC and TP53 mutation status (low AUC in ROC curves). The discussion offers plausible explanations, such as the DO-7 antibody detecting both wild-type and mutant protein and the potential for certain mutants to retain transcriptional activity. However, this central paradox needs a more nuanced and critical discussion. Recommendation: The authors should more explicitly discuss the different p53 IHC patterns recognized in the literature: (a) "Null" pattern (complete absence of staining, typically associated with truncating mutations), (b) "Overexpression" pattern (strong, diffuse nuclear staining in >50-60% of cells, often associated with missense mutations that stabilize the mutant protein), and (c) "Wild-type" pattern (variable, often heterogeneous staining). The finding that the "overexpression" pattern (≥50%) correlates with better survival is unusual and requires deeper exploration. Is it possible that this high expression represents a "hyper-stable" mutant p53 that, in the specific context of EGFR-driven oncogenesis, might still exert some residual tumor-suppressive function or alter dependency on EGFR signaling, as briefly mentioned? This should be the focus of the discussion.

  1. The "Protective Effect" of p53 Overexpression: The conclusion that p53 overexpression has a "protective effect" is a strong claim. Given that IHC cannot reliably distinguish wild-type from mutant p53, it is difficult to assert that the protein itself is protective. Recommendation: The language should be refined. It is more accurate to state that "the p53 IHC overexpression phenotype (≥50%) identifies a subgroup of patients with a more favorable prognosis," without definitively ascribing a protective function to the overexpressed protein itself. The alternative explanation—that this IHC pattern is a surrogate marker for a specific, less aggressive tumor biology—should be given equal weight.

  1. Sample Size and Statistical Power: The sample size (n=83) is modest, and the cohort is further subdivided for subgroup analyses (e.g., by EGFR subtype and TP53 mutation type). This increases the risk of Type II errors (missing true associations) and can make some subgroup findings appear less robust (e.g., the trend for Del19 in Figure 3b/f). Recommendation: The authors should explicitly acknowledge the limited power for subgroup analyses as a key limitation. For the survival curves in Figures 1, 3, and S2-S6, providing the number of patients at risk at relevant time points in the tables or figures would greatly help readers assess the reliability of the curves, especially towards the later time points where patient numbers drop.

  1. Methodological Limitations and Clinical Context:
  • Sanger Sequencing: The use of Sanger sequencing for TP53 (exons 5-8 only) is a significant limitation. It likely misses mutations in other exons and has lower sensitivity for detecting mutations in heterogeneous tumor samples compared to next-generation sequencing (NGS).
  • Lack of Osimertinib Data: As the authors note, the cohort was treated with first-generation TKIs (gefitinib/erlotinib) and not osimertinib, which is now the global standard of care for untreated EGFR-mutant NSCLC. This affects the clinical translatability of the findings.
  • Recommendation: These points are correctly mentioned in the limitations section but their impact on the study's conclusions should be emphasized more strongly in the discussion. The claim that p53 IHC "helps to spare patients from toxicity of novel combination therapy" (Simple Summary) is premature and should be tempered, as this study does not provide direct evidence for this in the context of modern TKI regimens.

Minor Points

  • Abstract: The abstract states "Tumor p53 overexpression (≥50%) was identified a strong prognostic factor..." It would be clearer to specify "a strong favorable independent prognostic factor for overall survival."
  • Results (Section 3, Paragraph 1): The description of the TP53 mutations is slightly confusing. It lists V216 deletions in exon 6 as "nondisruptive" and H178R in exon 5 as "disruptive." The rationale for this classification (referring to Molina-Vila et al.) should be briefly stated here or a reference to the Methods section made clearer.
  • Table 1: The categorization "WT+Disruptive MT" is unconventional and potentially misleading, as it groups wild-type with a subset of mutants. The rationale for this grouping should be justified in the table legend or methods.
  • Discussion: The sentence "Furthermore, mutated p53 may confer various gain-of-function activities... which, in turn, could make the tumor more sensitive to EGFR-TKIs" is highly speculative. While an interesting hypothesis, it should be framed as such and not presented as a likely explanation without supporting evidence from this or other studies.
  • Figures: The Kaplan-Meier curves in the main figures are clear. However, ensuring that all supplementary figures (S1-S6) are of publication quality and have clearly labeled axes and legends is crucial.

Conclusions and Recommendations

This manuscript presents a valuable and thought-provoking analysis that challenges the simple paradigm of p53 IHC as a mere surrogate for TP53 mutation status. The identification of a p53 IHC-high phenotype associated with improved OS is a noteworthy finding that deserves further investigation.

Please address the points above, particularly:

  1. Providing a more sophisticated discussion of the p53 IHC patterns and the potential reasons for the observed discordance and paradoxical prognostic finding.
  2. Tempering the language regarding the "protective" nature of p53 and refining it to describe the "prognostic association" of the IHC phenotype.
  3. More strongly acknowledging the limitations imposed by the sample size, Sanger sequencing methodology, and the historical cohort treated with first-generation TKIs.

Round 2

Reviewer 2 Report

Comments and Suggestions for Authors

The authors addressed the prior comments and I am pleased to accept their work.